# COVID-19 and Water Variables: Review and Scientometric Analysis

**DOI:** 10.3390/ijerph20020957

**Published:** 2023-01-05

**Authors:** Roxana Mare, Codruța Mare, Adriana Hadarean, Anca Hotupan, Tania Rus

**Affiliations:** 1Department of Building Services Engineering, Faculty of Building Services Engineering, Technical University of Cluj-Napoca, 128-130 21 Decembrie 1989 Blv., 400604 Cluj-Napoca, Romania; 2Department of Statistics-Forecasts-Mathematics, Faculty of Economics and Business Administration, Babes-Bolyai University, 58-60 Teodor Mihali Str., 400591 Cluj-Napoca, Romania; 3Interdisciplinary Centre for Data Science, Babes-Bolyai University, 68 Avram Iancu Str., 4th Floor, 400083 Cluj-Napoca, Romania

**Keywords:** wastewater, access to improved and safe drinking water, renewable water resources, freshwater withdrawal, scientific productivity, country, GDP per capita, HDI

## Abstract

COVID-19 has changed the world since 2020, and the field of water specifically, boosting scientific productivity (in terms of published articles). This paper focuses on the influence of COVID-19 on scientific productivity with respect to four water variables: (i) wastewater, (ii) renewable water resources, (iii) freshwater withdrawal, and (iv) access to improved and safe drinking water. The field’s literature was firstly reviewed, and then the maps were built, emphasizing the strong connections between COVID-19 and water-related variables. A total of 94 countries with publications that assess COVID-19 vs. water were considered and evaluated for how they clustered. The final step of the research shows that, on average, scientific productivity on the water topic was mostly conducted in countries with lower COVID-19 infection rates but higher development levels as represented by gross domestic product (GDP) per capita and the human development index (HDI). According to the statistical analysis, the water-related variables are highly significant, with positive coefficients. This validates that countries with higher water-related values conducted more research on the relationship with COVID-19. Wastewater and freshwater withdrawal had the highest impact on the scientific productivity with respect to COVID-19. Access to safe drinking water becomes insignificant in the presence of the development parameters.

## 1. Introduction

Water is both a primary resource for society and the crucial element in the prevention of COVID-19 (by WASH—Water, Sanitation, and Hygiene) [1]. It plays the main role in mitigating the spread of the disease, which is reaching its third pandemic year. As Larson [2] stated in his commentary, almost every infectious disease can be characterized by its relationship to water. This is the main reason why we chose to analyze the connection between this disease that caused a worldwide pandemic starting in 2020 and the interest presented to researchers in the field of water—which equates to the scientific productivity in the water field (in terms of published articles).

The water sector is extremely vast—it includes areas such as potable or freshwater, wastewater, water resources (surface water—rivers, lakes, and groundwater), recreational waters, marine waters, water management, water crises, floods, etc. Researchers tried to assess the relationship between the features of the COVID-19 pandemic and water in the different topics mentioned above.

Water management is the biggest and first factor responsible for the spread of COVID-19 and its eradication. To achieve good water management, especially at these crucial times, it is mandatory to discuss utilities, policies, policy responses, access to safe water for washing and drinking, water consumption, water treatment, water scarcity, and so on. First, each state should consider the human right to water, sanitation, and hygiene (WASH). Authorities must ensure access to safe water and sanitation and even turn to water shutoffs by enforcing water disconnection moratoriums, as in the case of some states in the U.S. [3]. This is an important measure considering increased household consumption during lockdown [4], especially in urban areas [5], and according to different patterns [6]. Unfortunately, not all countries around the world have permanent access to water. Many countries, mostly low-income or developing economies, are still facing water scarcity [7,8,9]. Over 1.6 billion people all over the world face “economic” water scarcity. This is due to a lack of necessary infrastructure [10], the depletion of renewable water resources (30% of the groundwater systems), and the increasing stress on the world’s main aquifers [11]. The quality of renewable water resources is extremely important due to their multiple uses in domestic plants, industrial and agricultural activities, land use, etc. The 2020 lockdown measures offered the possibility for a revival of most rivers and lakes affected by potential malfunctions of wastewater treatment plants (WWTPs) [12], industry—fishing and metals [13], urban pollution [14], agriculture, and land use [15].

Wastewater is a very important water-related variable. It is the first factor capable of providing information related to the disease through its composition. The particles of the COVID-19 enveloped virus can survive in wastewater [16]. Therefore, they need to be first detected in the wastewater [17,18], overseen [19,20,21,22] so they can be predicted [23,24], and then further removed from the wastewater [25,26].

Due to all the aspects presented above and the huge amount of information given by all water research topics, the present paper focuses only on the four most important water-related variables that influence human life and health. These water variables are: (i) wastewater, (ii) renewable water resources, (iii) freshwater withdrawal, and (iv) access to safe and improved drinking water. They are all directly or indirectly influenced by COVID-19. All these five interdependent factors are considered alongside the development parameters: GDP per capita (gross domestic product per capita) and HDI (human development index). The other reason we chose the present study is the lack of a similar kind of study in the literature. Many articles present reviews regarding COVID-19 and different topics related to one or two water topics, such as the WASH perspective in low-income countries [27]; water supply outages [28]; urban and rural water cycles [29]; and detection, survival, and disinfection technologies for COVID-19 [30], but none of them deepen a strict correlation analysis with COVID-19 and the four water variables considered together regarding scientific productivity. Moreover, no control/development parameters, such as GDP per capita and HDI, that express the development level of society were involved in any of the existing scientific papers. As for the bibliometric and scientometric parts, several studies were found, three of which debated the relationship between COVID-19 and scientific productivity in relation to the environment [31,32,33]. Another paper presents further research needs in the field of water, focusing on water-based epidemiology (WBE), WWTPs, the survival and detection of COVID-19 in the water cycle, and aqueous environments [34]. The last scientific paper reported the influence of sewer overflow on public health [35].

The goal of this paper is: (1) to present a critical review of the relationship between: COVID-19, wastewater, renewable water resources, freshwater withdrawal, and access to safe and improved drinking water; (2) to assess how scientific productivity (the number of published articles) in the field of water is determined by the incidence of COVID-19; (3) to assess how scientific productivity is influenced by the incidence of COVID-19 vs. the four most important water-related variables already mentioned above; and (4) to control for the stability of scientific productivity (published articles) vs. COVID-19 vs. the four water variables in the presence of the development level of each country based on GDP per capita and HDI.

The first step in achieving this goal is to create the database according to the scientific papers related to COVID-19 and its family and water found in two of the main important databases—Web of Science (WoS) and Scopus. The next step is to analyze this database according to the maps and networks developed between COVID-19 and water variables. The third step is to evaluate the influence of COVID-19 incidence at the country level upon the scientific productivity in the field of water on the one hand and of produced wastewater, freshwater withdrawal, renewable water resources, and access to safe drinking water on the other. To control for the stability of the results, we introduced the development level proxied by the GDP per capita and the human development index (HDI).

The methodological development of the research is described in Section 2. Section 3 provides a thorough critical review of the relationship between COVID-19 and water-related variables: wastewater, renewable water resources, freshwater withdrawal, and access to improved or safe drinking water, and how all these four topics developed during the COVID-19 pandemic. Section 4 presents and discusses the results of the econometric assessment of how COVID-19 and water-related factors impact scientific productivity in the field, while the last section provides conclusions.

## 2. Methodology

The scientific productivity of countries with respect to studies assessing the issue of water vs. COVID-19, SARS-CoV-2, and Coronavirus was evaluated using the two most important scientific databases, Web of Science (WoS) and Scopus. The two databases were consulted for articles published up to 2 March 2022, and double counting was avoided to ensure the consistency and quality of the information. A final sample of 853 articles was assessed for this research. The analyzed scientific papers were related to the four water variables considered in this study: (i) wastewater, (ii) renewable water resources, (iii) freshwater withdrawal, and (iv) access to improved or safe drinking water—essential for human health.

### 2.1. Relation between COVID-19 and Water-Related Variables

All valid results were first introduced in the VOSviewer program, a clustering and mapping tool for network data developed by Van Eck and Waltman [36,37]. This program is used to create and visualize maps and networks of different items: (i) the main keywords related to the aim and variables presented in Table 1 of this paper (the four water variables); and (ii) countries. All data results are analyzed from the point of view of co-occurrences, co-authorship, and links between them. Items are grouped into different clusters; they can contain one or more items, depending on the links and weight attributes of the item included in the network. The higher the weight of the item, the more prominently the item is visualized on the map [36,37]. Clusters located close to each other in the maps indicate close relations. All this mapping related to water variables and their relation to COVID-19 is further completed by a critical review of this part of the water field.

### 2.2. Statistic Analysis of the Determinants of Scientific Productivity at the Country Level

The second part of the analysis synthesizes all the information from the assessed articles and presents a country-level database with information related to the scientific productivity of countries on the idea of water vs. COVID-19. The entire statistical analysis was performed by using two specialized software programs: Tableau 2021.2 and STATA 15.

The dependent variable in our study is the total number of articles published per country on this topic. A final sample of 94 countries was used in the analysis (see maps in different figures). Variables and descriptive statistics are presented in Table 1.

High variations were depicted in the data through the descriptive analysis, with skewed distributions (Appendix A). Consequently, we transformed the variables using the natural logarithm. This reduces variation, making the distributions closer to the normal one (Appendix B), and allows for the ordinary least squares (OLS) estimation of possible non-linear relationships.

The second step of the analysis consists of cluster analysis. This method was used to group countries based on their features related to the study variables. As a first step, we cluster the countries in the sample based on their scientific productivity and COVID-19 incidence. In the second stage, we also include the water-related variables that we employ. ANOVA values are presented for each cluster stage, along with a description of the average values of the variables for countries in each cluster.

As previously stated, we employed the OLS estimation method to account for the impact of COVID-19 incidence and the water variables on scientific productivity by evaluating the relationship between them. Before the actual estimation, correlations were assessed in order to avoid multicollinearity, while the variance inflation test was applied post-estimation. As the water variables turned out to be highly correlated (Table 2), the OLS models were estimated with COVID-19 and each water variable at a time. Post-estimation tests were applied to check the quality of the models (results of the post-estimation procedures are available upon request). The stability and robustness of the results were addressed in two ways. First, we introduced the level of development proxied by two variables, GDP per cap (a more quantitative proxy) and HDI (a more qualitative proxy), to control for the stability of the relationships. Second, the regression models were tested for heteroskedasticity and afterward estimated in their robust form. The final results based on the robust estimations are presented and interpreted in this study.

## 3. Relationship between COVID-19 and Water-Related Variables

COVID-19 has continued to be the “star” since the world pandemic started at the beginning of 2020, even if more than two years have passed since then, with ups and downs in COVID-19 waves and versions. Alongside COVID-19 and its family (SARS-CoV-2, coronavirus, norovirus, COVID, and so on), the scientific production in the field of water encountered significant and important growth, especially because of the importance of water in the fight against this horrible disease—prevention and detection. Because of this, the four water-related variables—wastewater, renewable water resources, freshwater withdrawal, and access to improved or safe drinking water—are further separately analyzed with respect to their relationships and importance to COVID-19, respectively, and their impact on the scientific productivity in water science.

A general look at the global maps of co-occurrences and links (Figure 1) shows that the core of the maps is COVID-19, which is strongly connected to wastewater (completed by wastewater-based epidemiology and sewage) and water clusters. When discussing water, some of the researchers included everything in this general word, while others, depending on the theme of the papers, took separately the parts of water supply and quality, drinking water, hygiene, and public health, all included in renewable water resources (rivers and surface waters are clearly emphasized, especially in the Scopus map), freshwater withdrawal, and access to improved and safe water.

The greater volume of information on each water variable, such as wastewater, renewable water resources, freshwater withdrawal, and access to improved and safe drinking water, and their interdependence, determines further and deeper analysis of the influence of COVID-19 upon them and scientific productivity. It is inappropriate to analyze only one water-related variable without referring to and linking it to the other three given by the complete circle presented in Figure 2. All water variables are connected between them and further to COVID-19. As stated before, COVID-19 arrives in wastewater in different ways (WASH, urine, feces, etc.). Wastewater contaminates renewable water resources with effluents if the treatment process is not efficient. Water resources are the source for freshwater withdrawal; so, if water resources are infected, the water withdrawn is contaminated too. Therefore, drinking water can also be contaminated, which leads to the continuous spread of COVID-19 disease. Therefore, a vicious circle is created.

### 3.1. Wastewater

Wastewater is the most important theme and keyword encountered in our research (46% in Web of Science and 55% in Scopus). This has been translated into a great number of articles and a positive impact of wastewater on the scientific productivity linked to the current COVID-19 pandemic. This aspect is further validated by the regression results from Section 4, which clearly position wastewater in the first place regarding its impact on scientific productivity during the COVID-19 pandemic at the country level. The reason is represented by multiple actions that can and have to be taken in the fight against COVID-19, as presented in Table 3.

Wastewater is produced in both urban and rural areas, but in most countries, the main wastewater input is given by cities (sewage in rural areas is either absent or very little used). Moreover, Aquastat, the only database that provides full data that is closest to the present and appropriate for further analysis, refers to produced municipal wastewater. Moreover, the research referred to municipal wastewater and sewage [16], especially due to the presence of hospitals in cities [30,71,72,73] and not in rural areas. Zhou et al. [74] confirmed this aspect by presenting a survey on surveillance methods for COVID-19 in wastewater, but only for urban areas (clearly specifying at the end that further studies need to be conducted for peri-urban and rural areas). These are the reasons why the produced municipal wastewater is considered to be the first water variable in the analysis regarding the factors that conditioned the scientific productivity in the field of water during the COVID-19 pandemic. The Scopus map of wastewater co-occurrences in Figure 3 strengthens the approach by illustrating the presence of municipal wastewaters linked to sewage and wastewater in general. Wastewater is the leader of the third most important cluster after COVID-19 and its family in the WoS case, respectively, humans in the Scopus database. This emphasizes the strong relationship between the three most important elements: COVID-19 human infection reflects on wastewater. Sewage, another way of saying wastewater, must not be forgotten either. It is strongly connected with this water variable being included in the same cluster as wastewater (for Scopus) or in the next cluster as importance and dimension (in the WoS case). Sewage is responsible for the transport of the infected water with COVID-19 particles from the source to wastewater treatment plants and for possible further transmission and infection, especially in cases of breakdowns or malfunctions, and for WWTPs workers [75].

Wastewater is the main route of transmission of COVID-19 disease (as Figure 2 emphasizes) by contamination, which spreads its reach to the other water-related variables analyzed in this paper: water contamination [76], water quality, water purification [77,78,79,80,81], and renewable water resources (surface water and groundwater) [82,83].

### 3.2. Renewable Water Resources

Renewable water resources are divided into two groups: groundwater and surface water. In both cases, water resources are connected to COVID-19 or/and SARS-CoV-2, wastewater, drinking or tap water, water quality, and access by the population to improved and/or safe drinking water (in the Web of Science, Figure 4) further translates into human health (Scopus database, Figure 4). All these associations are normal and somehow intuitive. Water resources can be contaminated with COVID-19 particles by floods, effluents from WWTPs, fecal-oral transmission, industrial wastewaters, and sewage breakdowns. At the same time, renewable waters represent the sources of drinking water—tap and bottled water. As cleaner the water resources, as smaller the percentage of the spread of COVID-19. Therefore, surface water and groundwater quality are very important regarding scientific productivity during the COVID-19 pandemic. This aspect is also confirmed when discussing water resources and their relationships with the COVID-19 family and human health in the Scopus database (Figure 4). As the determinants will show in Section 4, the more water resources a country has, the more prolific its scientific productivity is.

Unfortunately, scientific research on groundwater resources emphasizes the diminishing quantity and quality of aquifers [84,85], such as in Nicaragua, for example, especially during the dry season [86]. So, water resources are affected by drought [87,88], rainfall or a lack of it, high temperatures, and increased evaporation, but also by human life and anthropogenic activities [89]. Industry wastewater has the biggest negative contribution to the quality of renewable water resources, followed by agriculture, land use, and commercial and domestic sewage production. To reduce the main polluting factors [85], new methods for pollution control and cleaning, especially of the rivers, were developed based on big data analyses [90], water quality assessment, the water pollution index [91], and magnetic solid-phase extraction (MSPE) [92]. Besides these, limited movement and factory closures led to environmental changes during the COVID-19 lockdown and pandemic [91,93]. All these good changes determined the increase in rainfall and, therefore, the improvement of the stressed aquifers [94]. Even so, considering that world and industry life restarted in 2021, it is of great importance to continuously monitor the changes in aquifers, rivers, and lakes.

### 3.3. Freshwater Withdrawal

At first glance, freshwater withdrawal appears to have the smallest influence on scientific productivity in the current COVID-19 pandemic when discussing the number of articles on this topic. There are no articles that debate only this topic in relation to COVID-19 (compared to the other water variables), and the percentage of scientific papers included in the database is extremely small (almost 5%). This is also confirmed by the low correlation coefficient of −0.229 in Table 2 between freshwater withdrawal and COVID-19. This means that freshwater is the least affected by COVID-19 (a perfectly normal aspect considering the water is filtered by rock, sand, and ground layers). Practically, according to the life cycle assessment, freshwater withdrawal should be the last option in ensuring the water demand in general [95] and drinking water demand for water savings, respectively.

On the contrary, the analysis of the determinants of scientific productivity at the national level in the presence of the development parameters reveals in, the second highest values for this water variable (a regression coefficient of 0.335, close to 0.366 for wastewater). This highlights the highest influence that freshwater withdrawal has upon the research output in the COVID-19 pandemic, along with wastewater, in the presence of the development parameters.

Freshwater withdrawal is strongly connected to renewable water resources [96,97] (relation strengthened by the highest correlation coefficients from Table 2: 0.661), and drinking water is affected by water insecurity, scarcity, and price [98,99].

### 3.4. Access to Safe and Improved Drinking Water

The spread of COVID-19 can be stopped by social distancing and the use of water—by washing and drinking safe and improved water provided by renewable water resources. Almost 30% of the global population has access to safe water, which is very low considering the importance of safe water to people’s health. Furthermore, more than 3 billion people lack basic, adequate access to WASH, especially in low- and middle-income countries [100,101,102,103], but this is true even for lower-income classes in high-income countries [11,99]. Of all categories, women and girls are the most vulnerable due to their position according to socio-cultural norms and responsibilities in the family [104]. Therefore, this water-related variable presented a great interest, as confirmed by the significant number of scientific papers published on this topic: 114 in WoS and 134 in Scopus. Safe drinking water is strongly connected to COVID-19 and SARS-CoV-2, an aspect strengthened by the statistical analysis and the results presented in Section 4.2, despite the fact that the World Health Organization indicates that this virus cannot be transmitted through drinking water due to the use of residual chlorine [102]. Water quality is in straight and tight relation to water supply from reservoirs and systems [105]. The quality of tap water is influenced by the change in drinking water demand during the COVID-19 pandemic. Furthermore, drinking water is connected to wastewater and sewage, WASH, and renewable water resources (represented in Figure 5 by surface waters and groundwaters). All these strong relationships are perfectly normal and intuitive. Drinking water can be easily contaminated with COVID-19 particles, especially when drinking it from public spigots, toilets, or any other water channel (i.e., canal) [106]. The correlation analysis from Table 2 also confirms the strongest links between access to improved or safe drinking water and COVID-19 and wastewater, respectively.

Water demand patterns radically changed with the pandemic; household consumption increased all over the world [4,5,6,107,108,109], while industrial and public (universities, colleges, schools, administrative buildings, hotels, etc.) sectors encountered a significant drop in consumption due to their temporary or permanent closures [110,111]. Therefore, the studies are focused on the problems that may occur in the water system infrastructure [112] due to the changes in water demand. Hence, the topics studied were the risks of the degradation of the water quality in buildings’ plumbing (increased levels of lead or copper) and the increased risks of the appearance and growth of bacteria counts (e.g., Legionella) [113,114,115,116,117].

An interesting aspect appears in the relationship between renewable water resources. A strong relationship should be expected between access to safe drinking water and water resources. Unfortunately, the correlation coefficient near zero in Table 2 shows an insignificant connection between the two water-related variables; this aspect is further confirmed in Section 4.2, when introducing the HDI control parameter.

## 4. Determinants of Scientific Productivity at the Country Level

### 4.1. Articles vs. COVID-19 Incidence

As we moved the analysis from the article to the country level, it was interesting to see not only the descriptive statistics of the variables (Table 1) but also the geographical positioning of water vs. COVID-19 scientific productivity.

Figure 6 presents:All countries with published articles in the fields of COVID-19 and water;All collaborations between different countries (namely, researchers from different countries) are based on the average number of publications per year in 2020, 2021, and the beginning of 2022. The bigger the bullet, the higher the number of articles and collaborations.

If the number of articles published during the three COVID-19 pandemic years is to be discussed, the countries with the greatest number of scientific papers were published mostly in 2021 and less in 2020, not to mention 2022, in which papers were only prevalent at the beginning. This reveals that researchers needed some time to become acquainted with this horrible and lethal disease, COVID-19, and to become accustomed, see what it is about, find ways of fighting against it, and try to see the positives.

In both databases, as seen in Figure 6, the USA is the leader [33] with the highest number of published articles (316). It is somehow to be expected considering its position and influence in the world and the high values for COVID-19 incidence and GDP per cap. All are translated into a higher interest in the topic of COVID-19 and water, more money spent on research, more researchers involved, and more connections with other countries (the USA collaborated with 62 other countries with respect to scientific productivity in the field of water during COVID-19 times, according to Figure 6). The USA (red in Figure 7) is followed by India (orange in Figure 7), China (salmon pink), the UK (salmon pink in Figure 7) (as seen in the Scopus database, Figure 6), and Australia (pink); all are followed by Spain (WoS database) in regards to published articles and country collaborations. The result is validated by Figure 7, which presents only the geographical distribution of the articles at the country level, considering all 94 countries taken into the study. The purple color in Figure 7 shows a small number of articles that decreases once with the color darkening, down to single articles where dark blue-colored countries are concerned.

While the USA also had a high COVID-19 incidence [118], it is not even in the top 20. The top five, based on the COVID-19 incidence rate, consists of Denmark, Slovenia, Iceland, Israel, and the Netherlands, with all but the latter two having a very low number of articles published on the topic (three from Denmark, three from Slovenia, and one from Iceland). The maps in Figure 7 and Figure 8 show that we have a mixture of both direct and reverse relationships between the COVID-19 infection rate and the scientific productivity in this field. This is quite interesting, as, logically, we would expect higher scientific productivity in countries with higher COVID-19 incidence rates.

Following the descriptive analysis results, clusters were constructed to see how countries in the sample group stacked up. Articles and COVID were first used as clustering variables and resulted in three clusters (see their components in Figure 9). The USA, UK, India, China, and Australia belong to the same cluster, with high average values for articles and medium for COVID. Table 4 supports the descriptive results. Cluster 3, formed by the above-mentioned countries, is characterized by the highest average center value for articles but middling values for the COVID-19 infection rate. Cluster 1 has the lowest average center value both for articles and for COVID, while Cluster 2 has the highest for COVID.

As the goal of our research is related to COVID-19 and water, the water variables were introduced into the analysis. In the first step, the boxplots for each cluster were constructed, and similar behaviors of wastewater, Renwres, and Freshwith (Figure 10) were observed. They all have very small values and variation for cluster 2, medium variation for cluster 1, and very high variation for cluster 3. With respect to Access, the lowest variation is to be found, once again, in cluster 2, with the highest in cluster 1.

### 4.2. Articles vs. COVID-19 vs. Water Variables

Finally, the water variables were included in the clustering procedure, and countries were now grouped into four clusters. Similar to the first step, the clustering diagnosis is presented in Table 4, and the clusters on the map are shown in Figure 11. The efficiency of the clustering procedure increases, as the BSS is much higher in the second step. Cluster 4 is made up of only Brazil, which has significantly higher values for wastewater and renewable water resources than any other country in the sample, with 44 articles published on the topic and quite a high COVID incidence (135.06) (Table 5). Cluster 3 is made up of China, India, and the USA, with the highest average center value for articles and high averages for the water variables with respect to the cluster centers. Australia, Argentina, Uruguay, Peru, Lebanon, Jordan, Israel, and most of the European countries belong to the second cluster. They are characterized by medium values for publications in the field, but the highest COVID-19 infection rates and the lowest values for wastewater, Renwres, and Freshwith. However, these countries have the highest access rates to safe-drinking water. The first cluster is made up of the least developed countries, except for a few, such as Canada and Russia. This group has the lowest values for articles and COVID, along with, as expected, the lowest share of the population with access to safe drinking water.

The major goal of our analysis is to evaluate whether the COVID-19 incidence and the water resources, wastewater production, freshwater features, or access to safe drinking water of a country impact the scientific production that links water to the current pandemic. The regression results presented in Table 6 clearly show that the influence of the COVID-19 incidence rate on the total number of articles published in this area is significant only when considered along with wastewater and freshwater withdrawal. The impact is positive—studies about the linkage between COVID-19 and water issues were, on average, conducted and mostly published in countries with high infection rates. The correlation analysis has pointed out that more developed countries had higher COVID-19 infection rates, and, usually, these countries have more money spent on research, so, once again, the positive sign is expected.

It is interesting that the significance of COVID-19 in Equation (3), when considered with freshwater withdrawal, disappears when the two development proxies are introduced in the regressions (Table 7, Equations (3.1) and (3.2)), whereas it becomes highly significant together when considered with renewable resources and access to safe drinking water. However, a very interesting result is actually the sign. In all cases, the coefficient of COVID (first value) is negative (Table 7).

Since the main goal of introducing the development proxies is to assess the stability of the relationship between scientific productivity in the field and COVID, on the one hand, and water variables, on the other, we can conclude that the relationship between COVID-19 infection rates and articles is not stable. This is somewhat shown by the second cluster analysis (Figure 11 and Table 5). We can see that the resulting clusters do not have the same behavior for all variables, just as described in the related part of the article.

The highest impact of COVID-19 on the research output appears in the models with access (Equations (4.1) and (4.2)).

With regard to the impact of water-related variables, it can be seen that their influence is positive and highly significant in all model specifications, with the exception of access to safe drinking water, which becomes insignificant when the development level is controlled for through the HDI. With respect to the control variables, just as expected, they have positive coefficients—higher scientific productivity with respect to water vs. COVID-19 is present in more developed countries. This can be explained by the fact that such countries have better research infrastructure and more money devoted to research and development. Both the GDP per cap and HDI are highly significant in all models.

Consequently, we can conclude that scientific production related to water vs. the pandemic was more prolific in countries with higher wastewater production, more renewable water resources, more freshwater withdrawal, and a higher share of the population with access to improved or safe drinking water. The USA is the leader, as stated before; India, China, and the UK complete the top four (as the number of published articles decreases from 316 for the USA to 130 for the UK). The ranking is almost the same as the one made at the beginning of 2021, even if more than one year of COVID-19 has passed since then. Important differences appear in the number of articles, which was significantly higher in 2022, and the exact order of each country on top, depending on the researched area [32,34]. Wastewater and freshwater withdrawal have the highest impact on scientific productivity (in the simple models in Table 5 and Table 6, with control variables in Table 7 too). Both variables have very similar coefficients of almost 0.4 (0.398 for wastewater and 0.373 for freshwater withdrawal) compared to the other two variables that reached low and very low coefficients (only 0.038 for access to safe and improved drinking water). On average, countries with higher values for these variables had higher scientific productivity. These results are also validated by the numbers given by the Aquastat database linked with the numbers of articles from our database: the USA is the first on top with an amount of 60.41 × 10^9^ m^3^/year of produced municipal wastewater. The USA is the leader in wastewater articles, with a total of 125 on this topic alone. A deviation appears in the cases of India and China. India comes in second, with 47 scientific papers, even if the produced wastewater is only 15.45 × 10^9^ m^3^/year, compared to China, which produces the second-highest amount of wastewater (48.51 × 10^9^ m^3^/year) and has only 34 papers on wastewater. Lastly is the UK, with a total of 37 articles on wastewater and 4.089 × 10^9^ m^3^/year of wastewater produced [38,39,40,41]. The highly impactful effect of wastewater is also emphasized in the density map in Figure 12 (VOSviewer).

When discussing freshwater withdrawal, Figure 13 presents the correlation between the scientific productivity at the country level for the top four countries and the data provided by Aquastat for this water variable [38,39,40,41]. India and China are the leaders in freshwater withdrawal, but with a significantly lower number of published articles compared to the USA. Of course, countries’ populations and surfaces must not be forgotten. Moreover, an important aspect is the life cycle assessment process and people’s awareness of saving water and naturally renewable resources. The lowest value of the total freshwater withdrawal for the UK shows this awareness of water savings among the English people (when freshwater withdrawal should be the last option). With all these, the UK has the same scientific productivity as China (which has a freshwater withdrawal value 70 times greater than the UK).

## 5. Conclusions

The highest number of published articles (in terms of the highest scientific productivity) between the beginning of the COVID-19 era, from the 2020s to 2 March 2022, in the Web of Science (WoS) and Scopus scientific databases confirm the interest in this topic of COVID-19 and water field. The literature review conducted emphasizes the high co-occurrence and interdependence of the terms COVID-19 and water-related variables: (i) wastewater, (ii) renewable water resources, (iii) freshwater withdrawal, and (iv) access to improved and safe drinking water. The interconnection of the relationships between water variables and COVID-19 disease is confirmed or rebutted by the control/development parameters: GDP per capita, and HDI. It is interesting to see how much and in what ways they influenced scientific productivity.

It was expected that countries with a higher COVID-19 incidence would be more interested in water management related to this disease and conduct more research. However, in fact, the situation is not quite the same. The USA, India, and the UK researched and published a lot on the subject but had medium COVID-19 rates. Interesting is the fact that countries with high COVID-19 rates (the top five) had low publication performance with respect to the water topic and vice versa. The research on water vs. COVID-19 was mostly conducted in countries with high values for the water-related variables that we considered, regardless of whether we used freshwater withdrawal with resources or wastewater. We can thus conclude that countries with more water resources, better access to water, and better wastewater management were more interested in finding out how they could manage the water issue at the time of the pandemic. The relationships remain stable in the presence of the control factors related to the development level. The latter group of variables also has a direct effect on scientific productivity. The relationship is not surprising, as more developed countries have more money devoted to research and a far better research infrastructure. With all these, the topic of COVID-19 and water still needs further research, since water secures our lives, health, and needs.

## Figures and Tables

**Figure 1 ijerph-20-00957-f001:**
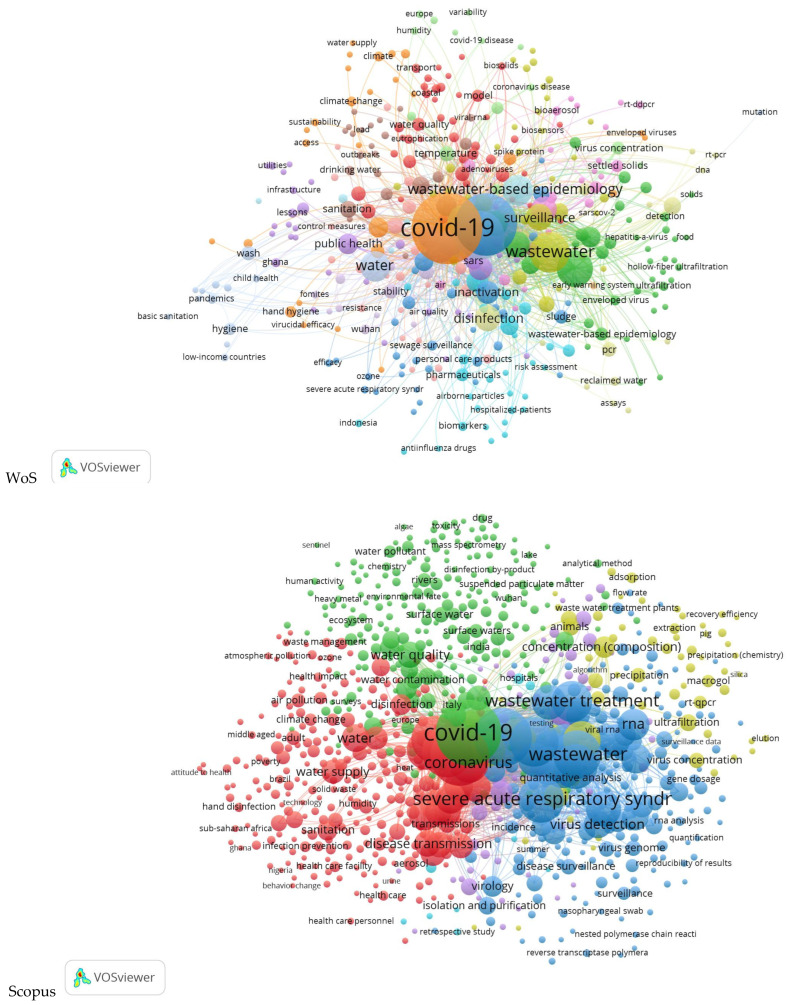
Global maps of co-occurrences and links for water and COVID for the two databases—WoS and Scopus.

**Figure 2 ijerph-20-00957-f002:**
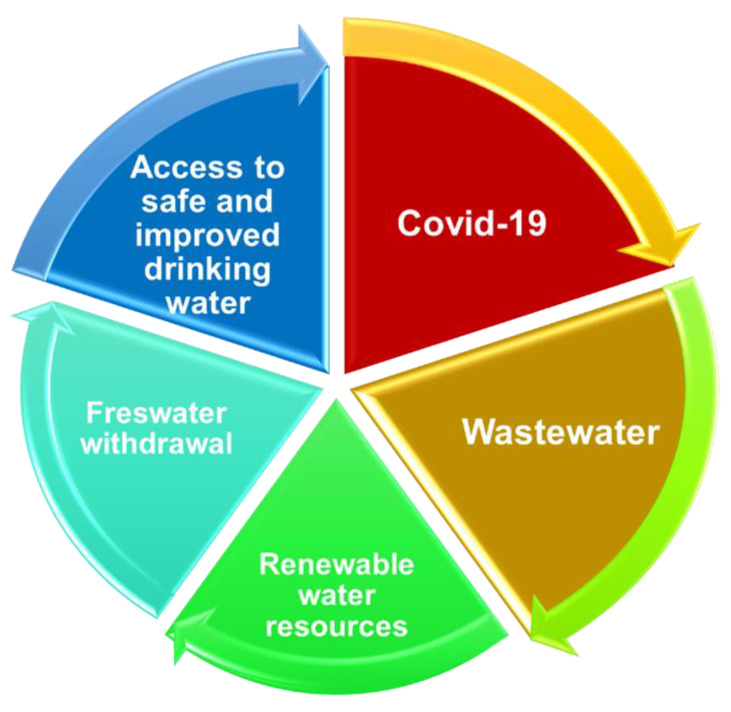
The interdependence of COVID-19 and water-related variables.

**Figure 3 ijerph-20-00957-f003:**
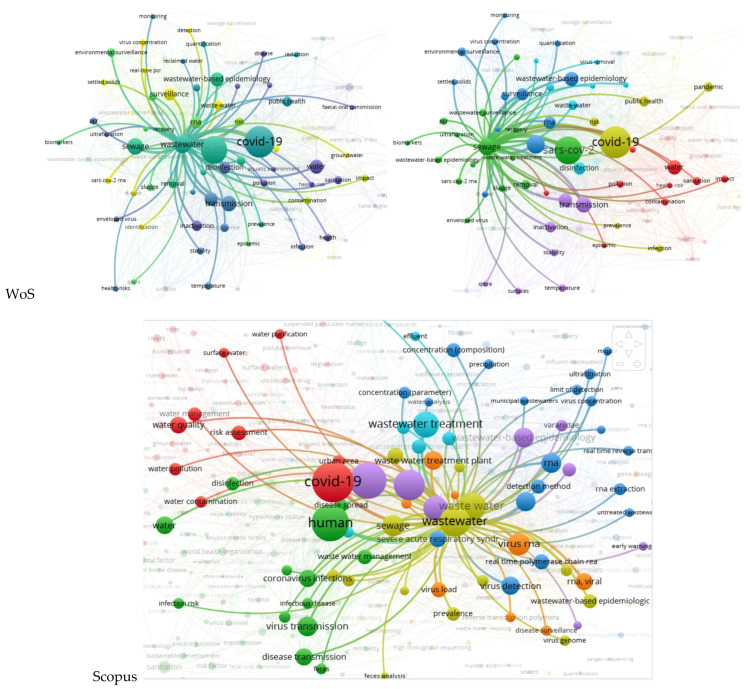
Co-occurrences for wastewater and sewage.

**Figure 4 ijerph-20-00957-f004:**
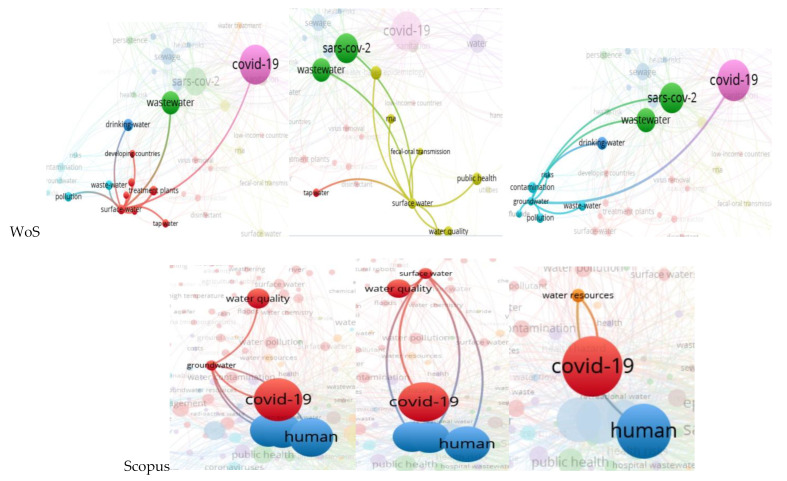
Relationships between renewable water resources (surface water and groundwater), water quality, humans, and COVID-19.

**Figure 5 ijerph-20-00957-f005:**
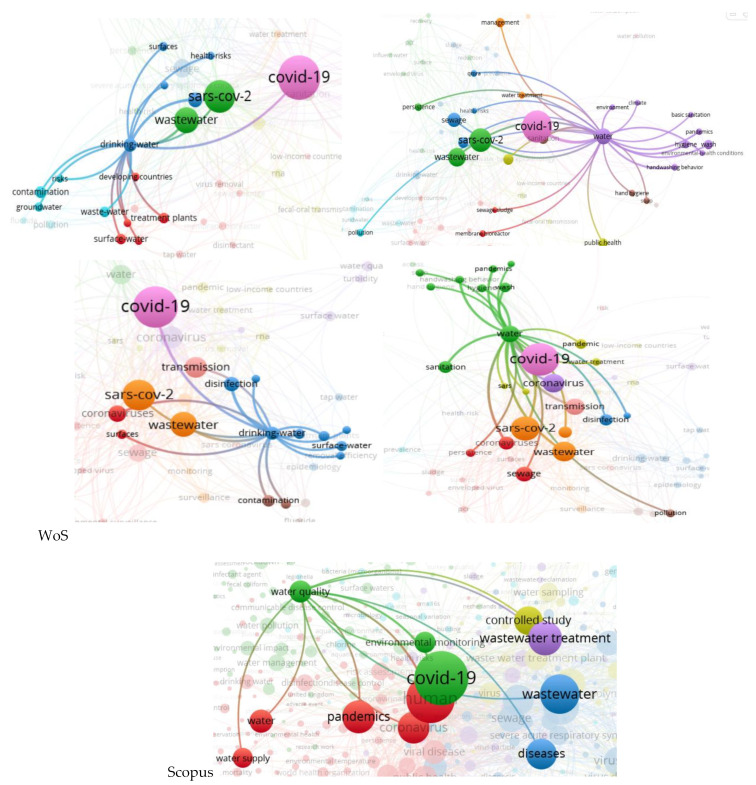
Co-occurrences between water, drinking water, water quality, COVID-19, and the rest of the variables (WoS and Scopus).

**Figure 6 ijerph-20-00957-f006:**
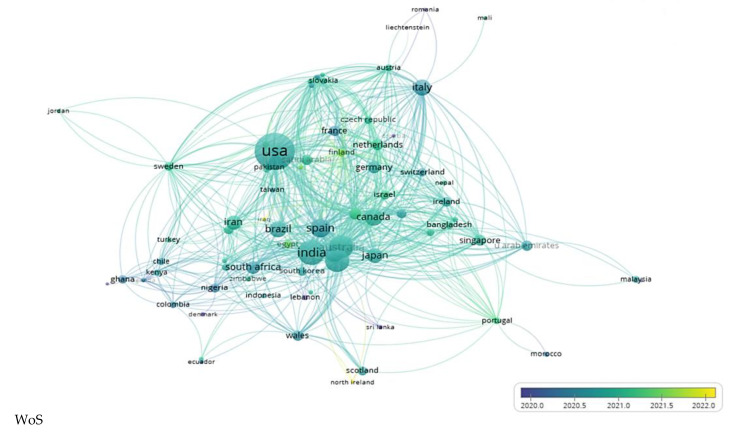
Countries’ coupling/collaboration based on the average number of publications per year in the WoS and Scopus databases.

**Figure 7 ijerph-20-00957-f007:**
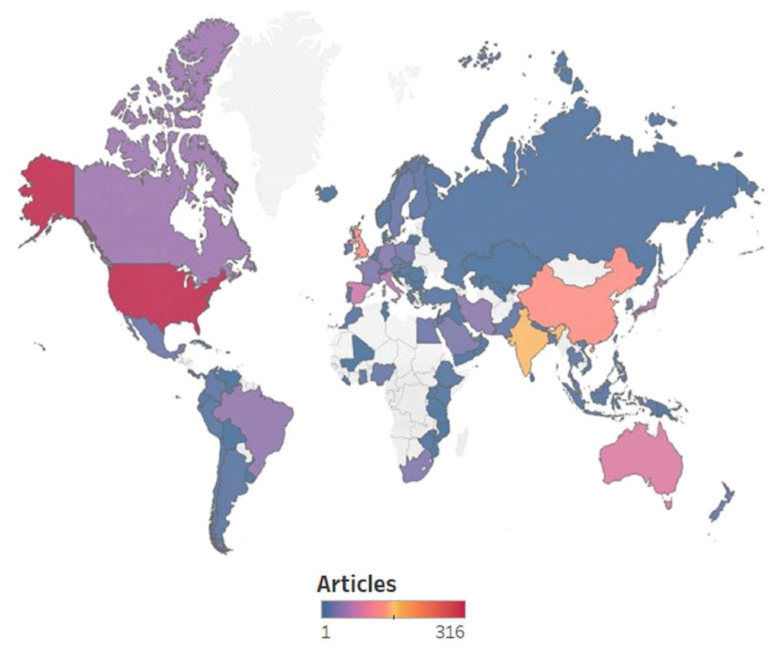
Geographical distribution of scientific productivity—articles.

**Figure 8 ijerph-20-00957-f008:**
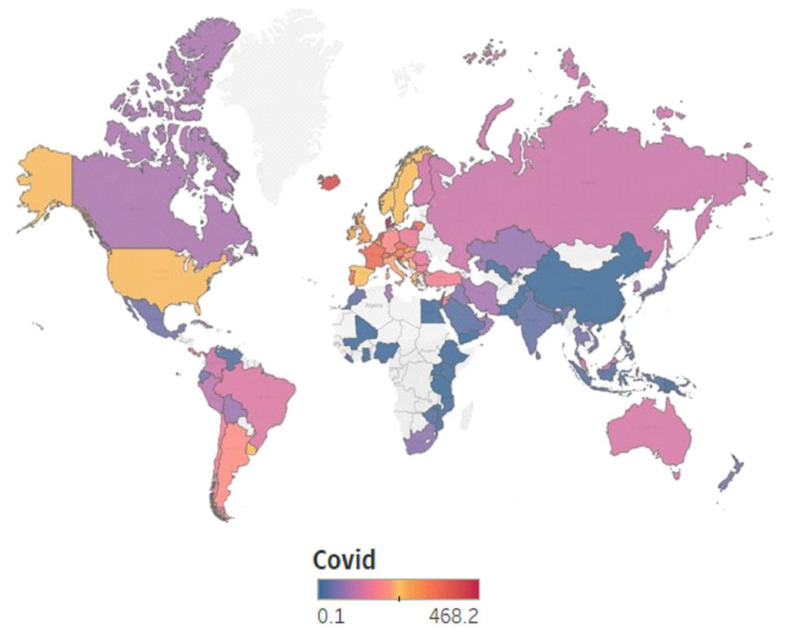
Geographical distribution of COVID-19 incidence—COVID.

**Figure 9 ijerph-20-00957-f009:**
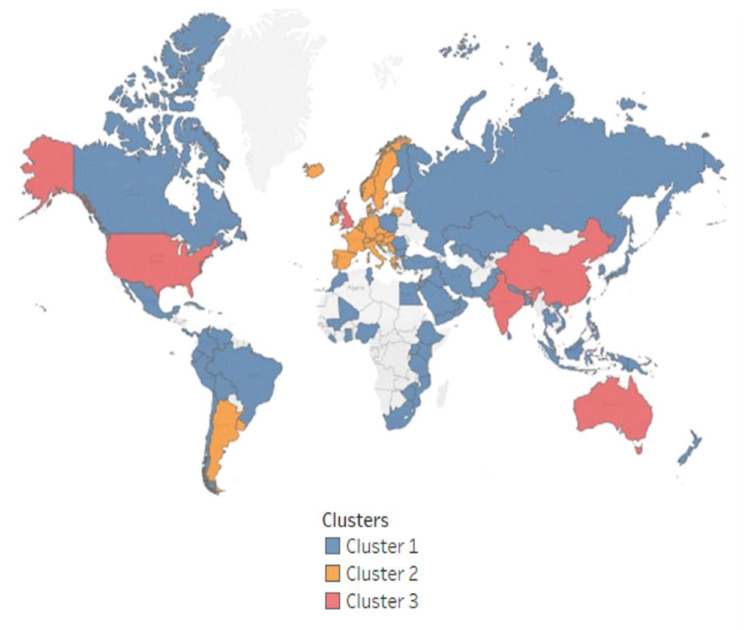
Cluster analysis—articles vs. COVID.

**Figure 10 ijerph-20-00957-f010:**
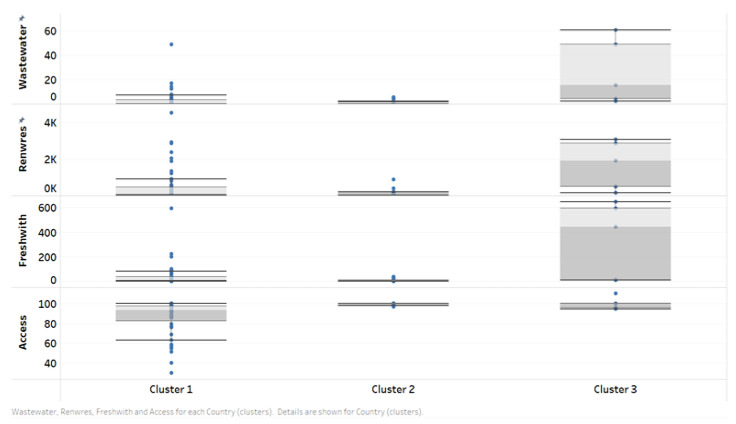
Water variables variation in the clusters based on articles and COVID.

**Figure 11 ijerph-20-00957-f011:**
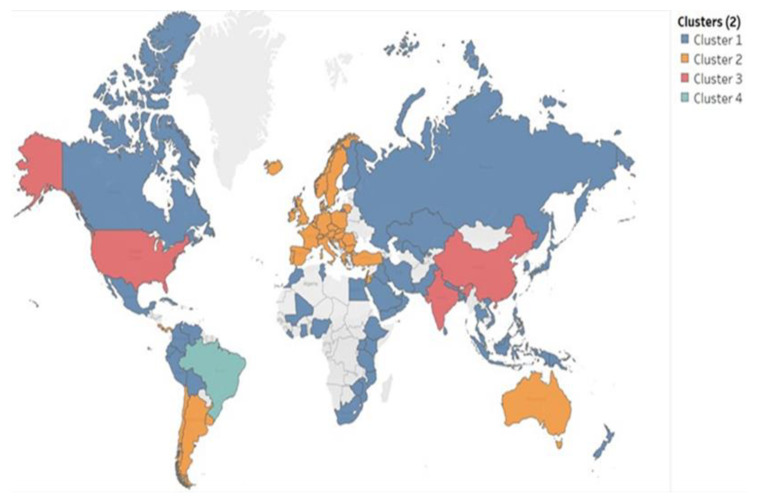
Cluster analysis—articles vs. COVID vs. water variables.

**Figure 12 ijerph-20-00957-f012:**
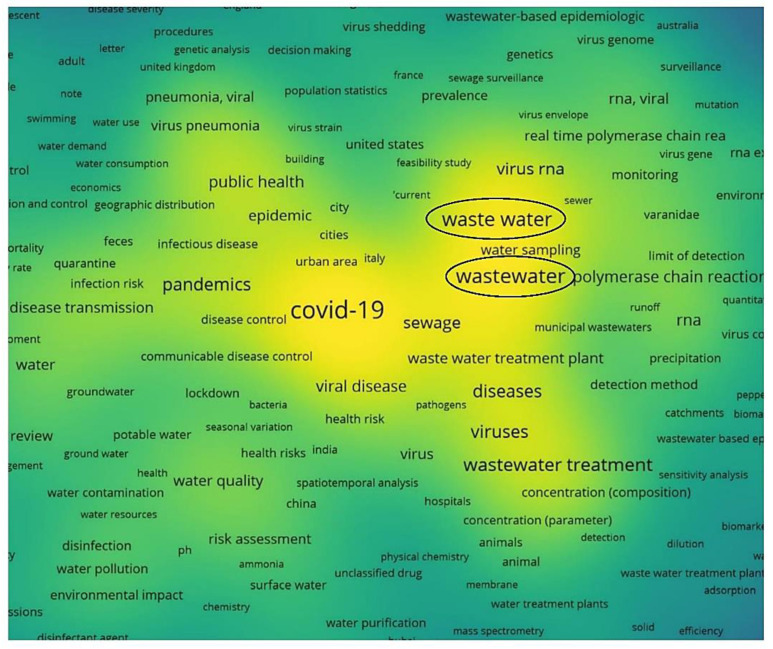
Density visualization map of the strongest relationship between COVID-19 and wastewater.

**Figure 13 ijerph-20-00957-f013:**
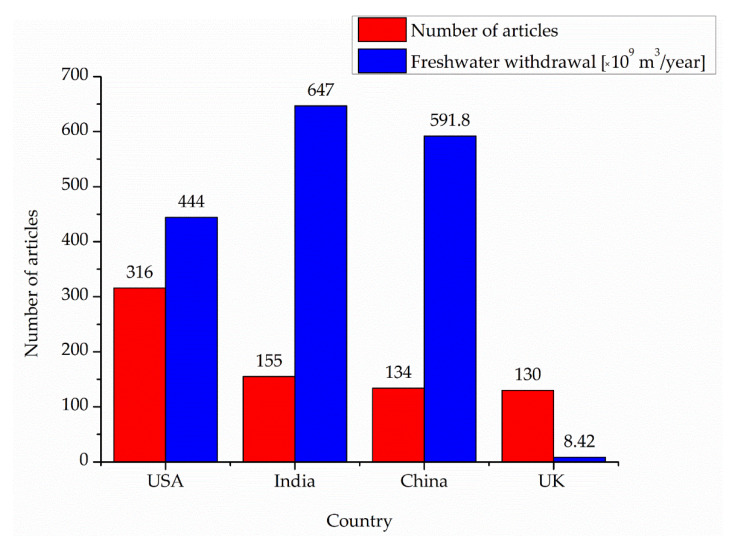
Correlation between scientific productivity and freshwater withdrawal for the top four countries.

**Table 1 ijerph-20-00957-t001:** Variables used in the analysis.

Variable	Description and Data Source	Min	Max	Mean	St. Dev
**Dependent**					
*Articles*	The total number of articles published per country. It represents the number of articles published by authors from a country on the Web of Science (WoS) or Scopus that have specific themes related to water vs. COVID-19. The articles were exhaustively included in the database, avoiding double counting for the ones present in both databases.	1	316	20.89	41.52
*Larticles*	Natural logarithm of Articles.	0	5.76	2.09	1.34
**Factors**					
*COVID*	COVID-19 incidence rate, computed by the authors as the total number of COVID-19 cases/1000 inhabitants in each country based on data provided by worldometers.info/coronavirus on 7 March 2022.	0.077	468.16	138.3	123
*LCOVID*	Natural logarithm of COVID.	−2.56	6.15	4.1	1.82
*Wastewater*	“Produced municipal wastewater (10^9^ m^3^/year). Produced municipal wastewater represents the annual volume of domestic, commercial and industrial effluents, and storm water runoff, generated within urban areas” ^1^ [38].	0.0006	7468	83.04	769.9
*Lwastewater*	Natural logarithm of Wastewater	−7.42	8.92	−0.29	2.2
*Renwres*	“Total renewable water resources (10^9^ m^3^/year). Total renewable water resources (TRWR) represent the sum of internal renewable water resources (IRWR) and external renewable water resources (ERWR). It corresponds to the maximum theoretical yearly amount of water available for a country at a given moment” ^1^ [39].	0	8647	520.1	1181.86
*Lrenwres*	Natural logarithm of Renwres	−2.85	9.06	4.28	2.61
*Freshwith*	“Total freshwater withdrawal (10^9^ m^3^/year) refers to the sum of surface water withdrawal, that is extracted from rivers, lakes and reservoirs, and groundwater withdrawal extracted from aquifers” ^1^ [40].	0.011	647.5	45.15	118.12
*Lfreshwith*	Natural logarithm of Freshwith	−4.51	6.47	1.89	2.19
*Access*	“Total population with access to improved or safe drinking water source (%). It represents the percentage of the total population using improved water sources. An “improved” source is one that is likely to provide “safe” water, such as a household connection, a borehole, etc. Current information does not allow yet to establish a relationship between access to safe water and access to improved sources, but WHO and UNICEF are examining this relationship. Safe drinking water is water that contains no biological or chemical pathogen at a level of concentration that is directly harmful to health. This includes treated, untreated, uncontaminated surface water, such as protected boreholes, springs and sanitary wells. The waters of rivers and lakes can only be considered healthy if water quality is regularly monitored and considered acceptable by public health officials. Reasonable access to water means a water supply in the water housing or within a 15-min walk of it” ^1^ [41].	30	110	92.16	14.26
**Control**					
*HDI*	“Human Development Index, computed by the United Nations. The Human Development Index (HDI) is a summary measure of average achievement in key dimensions of human development: a long and healthy life, being knowledgeable and have a decent standard of living. The HDI is the geometric mean of normalized indices for each of the three dimensions” ^2^ [42].	0.434	0.957	0.790	0.138
*GDP_cap*	“GDP per capita on 30 June 2021, according to World Bank. GDP per capita is gross domestic product divided by midyear population. GDP is the sum of gross value added by all resident producers in the economy plus any product taxes and minus any subsidies not included in the value of the product. It is calculated without making deductions for depreciation of fabricated assets or for depletion and degradation of natural resources. Data are in current U.S. dollars” ^3^ [43].	448.6	17,5813.9	21,581.8	26,537.3
*LGDPcap*	Natural logarithm of GDP_cap	6.11	12.08	9.2	1.39

^1^ Definitions provided by Aquastat, the database of the Food and Agriculture Organization of the United Nations. All data refer to the years 2017 and 2018, the most recent data available around the world. Most of the data have been preserved over time, which is why they can be accepted for the present time too [38,39,40,41]. ^2^ Definition provided by the United Nations [42]. ^3^ Definition given by the World Bank [43].

**Table 2 ijerph-20-00957-t002:** Correlation analysis.

	LCOVID	Lwastewater	Lrenwres	Lfreshwith	Access	LGDPcap	HDI
**LCOVID**	1						
**Lwastewater**	0.057 (0.5839)	1					
**Lrenwres**	−0.104 (0.322)	0.548 (0.000)	1				
**Lfreshwith**	−0.229 (0.027)	0.740 (0.000)	0.661 (0.000)	1			
**Access**	0.622 (0.000)	0.296 (0.004)	−0.009 (0.929)	0.120 (0.248)	1		
**LGDPcap**	0.684 (0.000)	0.204 (0.048)	−0.167 (0.109)	−0.152 (0.143)	0.672 (0.000)	1	
**HDI**	0.751 (0.000)	0.269 (0.009)	−0.114 (0.277)	−0.036 (0.731)	0.767 (0.000)	0.948 (0.000)	1

Corr. Coef. (*p*-value).

**Table 3 ijerph-20-00957-t003:** Actions in the fight against COVID-19.

Actions	Methods/Ways	References
Detection	✓ viral concentration techniques: polyethylene glycol (PEG) precipitation, ultrafiltration, electronegative membrane, ultracentrifugation	✓ Sangkham [44]; Torii et al. [45]; Ahmed et al. [46]; Wurtzer et al. [47]
	✓ Enzyme-linked immunosorbent assay (ELISA)	✓ Kilic et al. [48]
	✓ reverse transcription-polymerase chain reaction (RT-PCR)	✓ Ahmed et al. [24]; Heijnen et al. [49]
	✓ the reverse transcriptase quantitative polymerase chain reaction (RT-qPCR)	✓ Yaniv et al. [50]; Chik et al. [51]; Ahmed et al. [52]; Flood et al. [53]; Belhaouari et al. [54]; O’Brien et al. [55]; Hata et al. [56]
	✓ multiplex RT-qPCR	✓ Navarro et al. [57]
	✓ the reverse transcriptase droplet digital polymerase chain reaction RT-ddPCR	✓ Flood et al. [53]
	✓ next-generation sequencing (NGS), also known as high-throughput sequencing (HTS)	✓ Mackul’ak et al. [58]; Hui et al. [59]
	✓ paper-based diagnostic devices (PADs)	✓ Hui et al. [59]
	✓ nanoscale analytical tools and biosensors	✓ Mackul’ak et al. [58]; Yang et al. [60]; Bhalla et al. [61]
	✓ Rd-Rp based colorimetric reverse transcription loop-mediated isothermal amplification RT-LAMP	✓ Haque et al. [62]
Surveillance/monitoring	✓ clinical genomic surveillance	✓ Nag et al. [63]; Panchal et al. [64]
	✓ WBE surveillance for variant characterization	✓ Nag et al. [63];
	✓ RT-qPCR	✓ Zahedi et al. [65]; Bivins et al. [66]
	✓ WBE combined with artificial intelligence	✓ Randazzo et al. [67]; Abdeldayem et al. [68]
	✓ WBE based on biosensors	✓ Mackul’ak et al. [58]
Recovery of COVID-19 particles	✓ Concentrating pipette (CP)	✓ Ahmed et al. [69]
	✓ Adsorption-extraction (AE) method amended with MgCl2	✓ Ahmed et al. [69]
	✓ Ultrafiltration-based methods	✓ Flood et al. [53]
	✓ PEG precipitation	✓ Flood et al. [53]
	✓ Ultracentrifugation	✓ Wurtzer et al. [47]
Prevention	✓ WBE surveillance was completed by genome sequencing (NGS/HTS) and pathogen for variant characterization	✓ Nag et al. [63]; Panchal et al. [64]
	✓ WBE based on biosensors	✓ Mackul’ak et al. [58]
	✓ Tampon swab and RT-LAMP	✓ Bivins et al. [70]
	✓ RT-qPCR Diagnostic Panel	✓ Randazzo et al. [67]
Determination of microbial risks	✓ Quantitative assessment	✓ Hamadieh et al. [24]
	✓ Quantitative RT-qPCR	✓ Wurtzer et al. [47]; Flood et al. [53]; Hata et al. [56]
Determination of the real number of COVID-19 cases	✓ WBE combined with Artificial intelligence and their related Machine learning and Deep learning	✓ Ahmed et al. [24]; Abdeldayem et al. [68]

**Table 4 ijerph-20-00957-t004:** Cluster analysis diagnostics and features—articles vs. COVID.

Cluster	No. of Countries	Centers—Average	
		Articles	COVID
1	61	11.7	67.02
2	28	14.9	293.9
3	5	165.4	136.3
ANOVA			
BSS	5.62		
WSS	2.41		
TSS	8.04		

**Table 5 ijerph-20-00957-t005:** Cluster analysis diagnostics and features—articles vs. COVID.

Cluster	No. of Countries	Centers—Average					
		Articles	COVID	Wastewater	Renwres	Freshwith	Access
1	49	11.76	47.25	2.32	493.03	31.44	85.5
2	40	18.15	255.4	1.3	135.6	9.07	99.7
3	4	153	83.7	43.2	2665.1	568.9	97.2
4	1	44	135.1	7468	8647	65.7	98.1
ANOVA							
BSS	10.84						
WSS	5.98						
TSS	16.82						

**Table 6 ijerph-20-00957-t006:** Robust regression results.

Variables	Equation (1)	Equation (2)	Equation (3)	Equation (4)
**LCOVID**	0.029 (0.051)	0.096 (0.087)	0.159 ** (0.065)	−0.13 (0.103)
**Lwastewater**	0.398 *** (0.073)	-	-	-
**Lrenwres**	-	0.179 ** (0.051)	-	-
**Lfreshwith**	-	-	0.373 *** (0.05)	-
**Access**	-	-	-	0.038 *** (0.013)
**Constant**	2.09 *** (0.242)	0.961 *** (0.431)	0.737 ** (0.291)	−0.9 (0.936)
**R** ^2^	0.4305	0.1302	0.3557	0.1073
**F (Prob > F)**	15.15 (0.000)	7.03 (0.002)	30.92 (0.000)	5.09 (0.008)

Coef. *** followed by (robust std. error); ***, **, * denotes significance at 1%, 5%, and 10%.

**Table 7 ijerph-20-00957-t007:** Robust regression results—with control variables.

Variables	Equation (1.1)	Equation (1.2)	Equation (2.1)	Equation (2.2)	Equation (3.1)	Equation (3.2)	Equation (4.1)	Equation (4.2)
**LCOVID**	−0.128 * (0.066)	−0.128 (0.084)	−0.219 *** (0.068)	−0.252 *** (0.081)	−0.092 (0.072)	−0.085 (0.099)	−0.253 ** (0.099)	−0.285 ** (0.076)
**Lwastewater**	0.366 *** (0.071)	0.358 *** (0.072)	-	-	-	-	-	-
**Lrenwres**	-	-	0.211 *** (0.041)	0.191 *** (0.044)	-	-	-	-
**Lfreshwith**	-	-	-	-	0.371 *** (0.05)	0.335 *** (0.053)	-	-
**Access**	-	-	-	-	-	-	0.024 * (0.013)	0.016 (0.014)
**LGDPcap**	0.305 *** (0.112)	-	0.627 *** (0.112)	-	0.480 *** (0.103)	-	0.373 ** (0.167)	-
**HDI**	-	2.82 ** (1.262)	-	6.176 *** (1.161)	-	4.123 *** (1.248)	-	4.608 ** (1.976)
**Constant**	−0.083 (0.883)	0.494 (0.762)	−3.64 *** (0.917)	−2.546 ** (0.733)	−2.649 *** (0.768)	−1.451 ** (0.661)	−2.48 ** (1.09)	−1.899 ** (0.880)
**R** ^2^	0.4809	0.4628	0.3471	0.3096	0.4872	0.4304	0.1718	0.1721
**F (Prob > F)**	12.32 (0.000)	13.24 (0.000)	18.62 (0.000)	7.43 (0.000)	6.04 (0.000)	23.38 (0.000)	6.01 (0.000)	6.74 (0.000)

Coef. *** (robust std. error); ***, **, * denotes significance at 1%, 5%, 10%.

## Data Availability

Not applicable.

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
