# Peer review of "COVID-19 and Water Variables: Review and Scientometric Analysis"

_ijerph, 2023, doi:10.3390/ijerph20020957_

Round 1
Reviewer 1 Report
The material is very interesting. Reviewer didn't find any distrurbing information. The work is divided into 9 parts - which last are conclusions. it looks well organized. Data is well shown, and results are correct.
Author Response
Dear Reviewer,
Thank you very much for taking the time to read our work and send your valuable feedback. We appreciate your evaluation of our work, and we are grateful.
Reviewer 2 Report
Kindly find the attached file, the [reviewed] manuscript with all the comments. Thank you.

Author Response
Dear Reviewer 2,
Thank you for taking the time to read our work and send your valuable feedback that helps us improve it.
Please find our rebuttal to your comments in the pdf file and additionally an explanation regarding Tables 6 and 7 with the statistics. We hope we have succeeded in treating them in a satisfactory manner.
Tables 6 and 7: Dear reviewer, this is the standard format used in all research fields for the presentation of regression estimations. The output is very clear for any researcher, not confusing at all, as, in the footnote of both tables we stated: “Coef.*** (robust std. error);
***, **, * denotes significance at 1%, 5%, 10%”.
Thus, we consider it clear that the first value is the regression coefficient, followed by the star symbols according to the critical level employed for assessing the significance of the coefficient, which is described in the second line of the table’s footnote. Into brackets, we present robust standard errors. We do not present the probability (also known as the p-value), but we present the robust standard error, which is used to compute the test statistic and obtain the p-value. This is, once again, a custom in any data analysis procedure, as in this way, any reader can actually check the accuracy of our results. We do not have what to change in tables 6 and 7, as any change would be against the statistical and econometrics principles.

Reviewer 3 Report
The problem of COVID and it's influence all spheres of our life is an interesting case of study. The literature review covered a fairly large layer of articles concerning some aspects of water problems. Work has classical structure. The article is quite detailed, with a large amount of analyzed information.
Author Response
Dear Reviewer 3,
Thank you for taking the time to read our work and send your valuable feedback. We appreciate your evaluation of our work, and we are grateful.
Reviewer 4 Report
Covid-19 and water variables: review and scientometric analysis
We are in front of a new review very rich in useful information, it can be accepted after the following revisions:
It was necessary to put the full names of the abbreviations GDP and HDI in the abstract.
· Please keep the same style of citing articles throughout the manuscript.
· Line 178 : Statistical analyses were conducted in Tableau 2021.2 and STATA 15, I did not understand this sentence.
· Fig 1 : Please specify the two databases in this figure.
· Please show insights from this review and how this work can be useful for researchers and scientists
· Can we know whether the references cited and used in this study are open access or not and how the authors can manage this enormous number of references.
· Line 518- 527 : it was better to represent the data in the form of histograms.
Author Response
Dear Reviewer 4,
Thank you for taking the time to read our work and send your valuable feedback that helps us improve it.
Please find, our rebuttal to your comments in the attached file. We hope we have succeeded in treating them in a satisfactory manner.

Reviewer 5 Report
This study is good and useful because it keeps pace with the event and it is important in terms of the following: This pathogen infected the world, but there are some comments on this study, for example:
1. The English language is shallow, it needs more review in order to understand the intended meaning.
2. Explain the direct health impact of Covid-19 on water, as shown in previous studies in terms of microbes or related diseases.
3. Figures No. 6 and 7 need more clarification in order to be understood.
4. Please set the chemical and standard one in each review.
5. Define abbreviations such as GDP, HDI, at the beginning of the review, and in the abstract.
Author Response
Dear Reviewer 5,
Thank you for taking the time to read our work and send your valuable feedback that helps us improve it.
Please find our rebuttal to your comments in the attached file. We hope we have succeeded in treating them in a satisfactory manner.

Round 2
Reviewer 2 Report
The comments are provided for amendment, and the authors need to revise the manuscript according to ALL the comments. Effort only made to answer at the authors' respond.
Author Response
Dear Reviewer,
Please find the revised form of the manuscript with responses to your observations. They are highlighted with comments.
Thank you!
